# Urban Densification Effect on Micrometeorology in Santiago, Chile: A Comparative Study Based on Chaos Theory

**Patricio Pacheco** [1,*] , **Eduardo Mera** [1,*] **and Giovanni Salini** [2]

1 Departamento de Física, Facultad de Ciencias Naturales, Matemáticas y Medio Ambiente, Universidad Tecnológica Metropolitana, Las Palmeras 3360, Ñuñoa, Santiago 7750000, Chile
2 Departamento de Matemática y Física Aplicadas, Facultad de Ingeniería, Universidad Católica de la Santísima Concepción, Alonso de Rivera 2850, Concepción 4090541, Chile; gsalini@ucsc.cl
* Correspondence: patricio.pacheco@utem.cl (P.P.); emera@utem.cl (E.M.)

**Abstract:** The concentration distribution of anthropocentric pollutants is favored by urban densification, affecting the micrometeorology in big cities. To examine this condition, chaos theory was applied to time series of measurements of urban meteorology and pollutants of six communes of the Metropolitan Region of Santiago de Chile, in two periods: 2010–2013 and 2017–2020. Each commune contributes, per period, six different time series: three for the meteorological variables (temperature, relative humidity, and magnitude wind speed) and three for the atmospheric pollutant concentrations ($PM_{10}$, $PM_{2.5}$, and CO). This qualitative study corroborates that each of the time series is chaotic through the calculation of chaotic parameters: Lyapunov exponent, correlation dimension, Hurst coefficient, correlation entropy, Lempel–Ziv complexity and fractal dimension. The variation in the chaotic parameters between the two periods can be interpreted in relation to the roughness change due to urban densification. More specific parameters, constructed from the Kolmogorov entropies and the fractal dimensions of the time series, show modifications due to the increase in the built surface in the most current period. This variation also extends to micrometeorology, as is clear from the Lempel–Ziv complexity and the Hurst coefficient. The qualitative picture constructed using chaos theory reveals that human interaction with nature affects diversity and sustainability and generates irreversible processes.

**Keywords:** atmospheric boundary layers; dissipative systems; entropy; urban roughness

## 1. Introduction

### 1.1. Micrometeorology

Micrometeorology is the area of meteorology that deals with observations and processes on smaller scales of space and time [1], in the order of 1 km and measurement periods of one hour or less. These processes are limited to the lower part of the planetary boundary layer, known as the atmospheric boundary layer (ABL) [2]. Its foundations are determined by the exchange of energy, mass, and gasses between the atmosphere and the base surface (water, soil, plants). The surface of the earth is a boundary that greatly influences the atmosphere, especially in the air properties of the ABL, where the effect of friction and the thermal effects of heating and cooling of the surface trigger considerable flux, that transport momentum, heat, humidity, or matter [3]. The ABL has different atmospheric regimes that respond to its sublayer structure. This structure experiences an evolution that is parallel to the diurnal cycle. Separating the free atmosphere from the mixed layer is a strongly stable entrainment zone of intermittent turbulence. At night, turbulence in the entrainment zone ceases, leaving a nonturbulent layer called capping inversion that remains strongly stable [3].

The study of temperature and wind (turbulence) being affected by external factors, such as buildings [4], flora, population, and relief, is of paramount relevance. International networks of flow control sites use micrometeorological techniques to understand the exchange of energy and mass between the biosphere and the atmosphere [2]. On the one hand, heat transfer through the boundary layer firstly occurs by molecular diffusion and then by turbulent diffusion [4]. In such environments, the flow structure in the roughness sublayer dictates the air flow and pollutant dispersion within and above urban canopies [3,5]. This sublayer consists of the lowest part of the atmospheric surface layer, from the ground to 2–5 times the average height of the canopy elements. Moreover, the roughness sublayer is where the turbulent exchange of momentum, heat, and mass occur [5,6]. On the other hand, turbulence enables the exchange of $CO_2$ between plants and the atmosphere, whilst also aiding the distribution of pollen. Finally, buildings play a crucial role within micrometeorology, as they increase turbulence and thermal sensation in cities [1,4].

From a historical perspective, stationary and mobile studies of micrometeorological changes were conducted in the nascent city of Columbia, Maryland, in late 1967. Measurements showed the development, intensification, and expansion of an urban heat island. This phenomenon is essentially attributed to the disruption of heat flow in and out of the ground caused by the change in surface characteristics. In this sense, approximately half of the observed change in relative humidity could be allocated to a decrease in evapotranspiration when replacing vegetation with concrete and asphalt [7,8].

Saaroni [9] devised a method to estimate the formation of a net urban heat island (UHI) in the canopy layer in regions with complex configuration grounds and with no preurban observations. As a test, the UHI procedure was applied to an arid city, Beer Sheba, Israel, for minimum and maximum air temperatures in the summer and winter seasons. With these characteristics, the UHI's estimated net contribution in Beer Sheba ranges from +0.8 °C to +3.1 °C, with high positive values during night hours, showing a good fit with previous studies. This implies a further intensification of heat stress during the summer beyond the historic base level, which is expected to increase in the region [9].

Ref. [10] Shows the influence of land use against the phenomenon of urban heat island (UHI) in Jakarta, Indonesia, using different types of urban land use within simulations carried out with WRF (Weather Research and Forecasting Model). Here, it follows that the variation between percentages of portions of land used with buildings and vegetation is important in making decisions about the future planning of the city.

Having urban input data that are as up-to-date and robust as possible allows us to adjust the simulations, and thus, improve results. Works such as [11] compare the results obtained previously and new estimates with data generated from different sources that were not previously available.

More recent studies make estimates of thermal conditions and urban winds in digital models of buildings [12]. This study confirmed that comparison of microclimate thermal conditions based on measurements and obtained from modeling using SkyHelios are in sufficient agreement and can be used in urban planning in the future [12].

In Chile, there is an online public network for monitoring meteorological and pollutant variables [13] that has extensive periods of data records that form time series. The field recording of data serves as a starting line for this research. To analyze the data, we followed the line of chaos theory applied to time series. Many authors have approached the treatment of time series in this way [14–18]. From the extensive data records, measurements of meteorological variables (magnitude of wind speed, relative humidity, temperature) and air pollutants (carbon monoxide, $PM_{2.5}$, and $PM_{10}$ particulate matter) constituted in time series for two periods 2010–2013 and 2017–2020 were selected in the first approximation. The analysis of these series, through a software that explores their chaoticity, allows us to determine if their nature is chaotic from the ranges of five parameters characteristic of chaos theory: Lyapunov's coefficient ($\lambda > 0$), Hurst's coefficient ($0.5 < H < 1$), correlation dimension (Dc < 5), correlation entropy ($S_K > 0$), and Lempel–Ziv complexity (LZ > 0).

Finally, the comparison between the chaotic parameters of the meteorological variables and of the pollutants allows us to analyze their variation by commune, between study periods, and due to the built area (in m²) and the increase in tall buildings [19,20].

### 1.2. Urban Densification

Many cities in Chile have urban growth that is becoming unsustainable, especially in densely populated metropolitan areas. Informal settlements grow, covering plains and mountain ranges, to produce very problematic cities whose exponential growth puts public services under pressures that they are ill-equipped to deal with. This model city, in addition to promoting the use of fuels, favors the urbanization of areas that provide environmental services, inducing exclusion dynamics between the city centers and the peripheries. Densification is a public policy concept that has been promoted by governments, experts, and international agencies as a solution to the problem of city dispersion. To densify is to use urban land more intensively. Some actions that characterize the densification plans are the construction of vertical houses, including the conversion of so-called underutilized areas, which are a common issue in many cities [21,22].

This study uses public data [19,20] to analyze urban density and the change in roughness of selected places. From a housing approach, the comparative study is important. According to official data [19,20], the number of residential properties (N°) between the years (t) 2009–2020 grew exponentially: $N° = 3.32 * 10^{-14} e^{0.0226[\frac{1}{Years}]t}$. By 2015, many of the buildings were 15 floors, and in 2020 the majority were 30 floors. The supply of apartments (2020) increased by 77.3%. In the Metropolitan Region, there were 50 homes/hectare in 1990; in 2020 there were around 5000 homes/hectare [19,20]. According to census data from 1970, the proportion of apartments in Chile was only 7% (116,748). Since then, that alternative has been widely more accepted by Chileans, so that in 2002 that percentage reached 12.6% (474,199, according to the 2002 census). In 2018, they represented 17.5% of a total of 1,138,062 homes (552,678 more than in 2002) [19,23,24].

In the following years, this verticalization process continued intensively through buildings made of concrete. The monitoring stations used in this study (see Figure 1) are located in areas that have gone through an intense process of high-rise building.

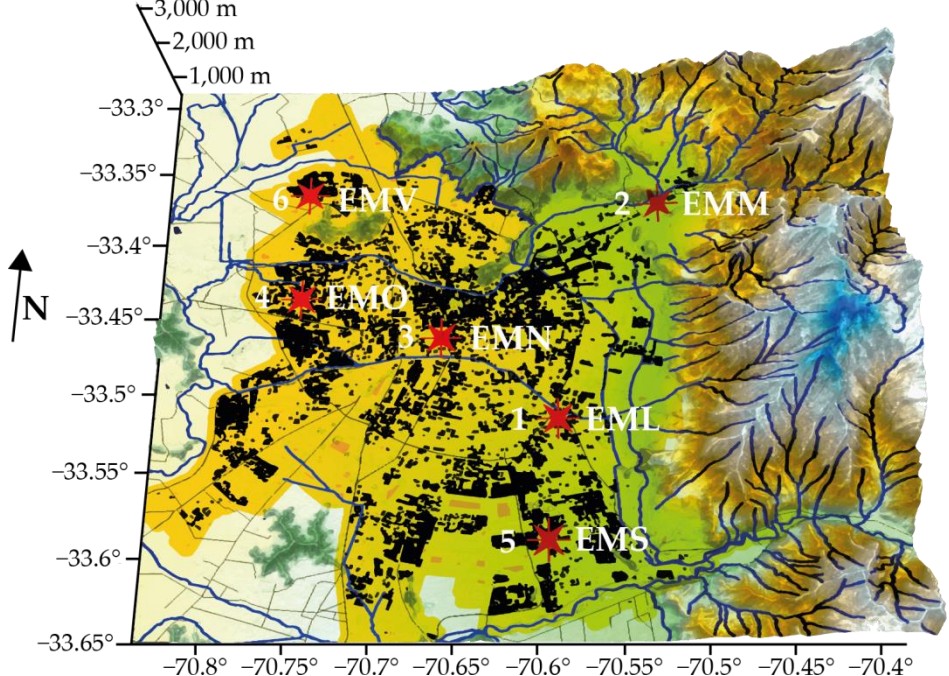

**Figure 1.** Geographical distribution of the monitoring stations for this study (red stars) and population density (dark areas).

For the crop canopy, the average in the metropolitan region in 2017 was 6.4 m$^2$/inhabitant, in 2021 it was 6.2 m$^2$/inhabitant, while the minimum value recommended by WHO (World Health Organization) is 9 to 11 m$^2$/inhabitant. The forest cover of the communes studied is as follows (approximate data, [19]): Las Condes (EMM): 37.3%, Puente Alto (EMS): 20.3%, La Florida (EML): 17.6%, Santiago-Parque O'Higgins (EMN): 11.4%, Quilicura (EMV): 7.7%, Pudahuel (EMO): 4.3%.

From the public database [19,20], it was possible to obtain the square-meter areas built for the two periods under study. In standard roads, constructions and coatings, concrete is extensively used due to its firmness, low cost, good performance in the face of wind loads and fire resistance. How does this type of material interact with the environment? Albedo is a dimensional magnitude that gives the percentage (from 0%, black body, to 100%, white body) of radiation that any surface reflects relative to the radiation that affects it. Typical albedos of various surfaces are [25], in % (albedo), e (emissivity, 20–25 °C): walls—concrete (10–30%; 0.94), walls—stone (20–40%; 0.85–0.95), fresh–old asphalt (5–20%, 0.95), stone (20–30%, 0.67), soil—sandy (20–25%, 0.91), urban medium (15%, 0.98), buildings (9%, 0.95).

From the data presented, it appears that massive urban densification using current construction materials generates heat gradients that alter the initial conditions of the boundary layer in which it is located [26]. That is, an alteration of the heat flow in and out of the ground is caused by the change in surface characteristics [8]. In addition to the change in surface roughness, this heat disruption affects natural wind regimes, making them more turbulent. These effects, fueled by a continuous construction process, can contribute to creating an artificial dissipative system that alters local weather conditions by inducing irreversible processes, just as contaminants do over the atmosphere [17,24]. Some countries have been concerned about this issue by promoting research into old (wood revaluation) and new materials for construction [27]. These have the potential to replace petroleum-based plastics, in line with the concept of a zero-waste policy.

## 2. Theoretical Perspective

### 2.1. Dissipation and Complex Systems

According to Prigogine [28], nonequilibrium is the creator of so-called dissipative structures, since they only exist far from equilibrium, requiring a certain dissipation of energy for their survival, and therefore, the maintenance of an interaction with the outside. Analogous to that of a city, which exists only in terms of its operation and maintenance of exchanges with the outside, a dissipative structure disappears when it ceases to be "fed". It is precisely this concept introduced by Prigogine which serves as the basis for one of the working hypotheses: pollutants would behave analogously to a dissipative structure, being fed by human beings, when emitting particles and other pollutants into the atmosphere. This is a situation that would not be reversed if it were not for the interaction generated by the meteorological parameters present in the Earth–atmosphere system [17].

The existence of effects generated by the interaction between urban areas and the atmosphere makes urbanization a topic of interest. These effects cause various phenomena, such as temperature increases and changes in turbulence within cities, which have an impact both directly and indirectly on people's quality of life. Characteristics such as the geometry of the city and the change in land use it generates are aspects that cause changes in the flows of gases from the atmosphere. The time series that make up the set of measurements of meteorological variables and pollutants, on which this work is based, were made within the urban boundary layer at a maximum height of 10 m above the ground (they are under the maximum range of up to 2–5 times the average heights of the canopy), which is a layer of great turbulence.

Turbulence is a property of flow and has some characteristic properties such as irregularity, three-dimensionality, diffusivity, dissipation, and high Reynolds numbers. Various investigations seek to improve the turbulence closure formalism through one-dimensional multilayer urban canopy models: large eddies are analyzed through numerical simulations in a variety of idealized geometries and flow regimes. The study reveals a good charac-

terization in the declared boundary conditions [29]. Other investigations have carried out wind tunnel experiments [30] in four urban morphologies: two tall canopies with uniform height and two super tall canopies with a great variation in the height of the elements (where the maximum height of the element is more than double the average height of the canopy, $h_{max}$ = 2.5 $h_{avg}$). The results point towards the existence of a constant tension layer for all the surfaces considered, despite the severity of the surface roughness. What is the difference with the previous literature? Lagrangian stochastic models are widely applied to predict and analyze turbulent dispersion in complex environments: land canopy flows and marine flows. The lack of empirical data does not allow us to understand how the particular characteristics of highly heterogeneous canopy flows affect Lagrange statistics. Researchers [31] use this formalism and Lagrange statistics in small time frames (thousandths of a second) to analyze empirical Lagrange trajectories in subvolumes of space that are small compared to canopy height. In their study, they use 3D Lagrangian trajectories measured according to a dense canopy flow model in a wind tunnel, using an extended version of real-time 3D particle tracking velocimetry. A key result obtained was that the random turbulent fluctuations, due to the intense dissipation, were shown to be more dominant than the inhomogeneity of the flow, affecting the Lagrange statistics in a short time. This is equivalent to the quasi-homogeneous regime of small-scale Lagrangian statistics.

When fluids are in a turbulent state it is dissipative, since viscosity, at the smaller scales, dissipates the kinetic energy they receive in a cascade from the larger scales, as indicated by Kolmogorov's theory [32]. Namely, an energy cascade results from the fragmentation of swirling structures that decompose successively into smaller eddies when the fluid is in a turbulent state. The idea that energy is cascaded from the large scales to the smaller ones, where it dissipates, was proposed by Richardson [33], and the theoretical formulation was Kolmogorov's [32]. The concept of scale was unclear and was interpreted as the wave number by Onsager [34], Weizsacker [35] and Heisenberg [36]. However, numerical simulations do not show Kolmogorov's cascade when using the wave number as a scale. Through a new definition of scale, based on the low-pass filtering velocity field, the existence of the cascade was confirmed [37].

Kolmogorov entropy is related, through the cascade of turbulence, to heat as a form of dissipation. This is what drives the dispersion of pollutants away from a deterministic process [38]. On the small scales, near the surface, the finer eddies disperse their energy as heat due to viscosity. So, the fully developed turbulent medium is characterized by two quantities, the average energy dissipation rate, $\varepsilon_D$, and the kinematic viscosity $\nu$. The dimensionality of $\varepsilon_D$ is energy/time/mass, $L^2 T^{-3}$, and the dimensionality of $\nu$ is $L^2 T^{-1}$. By combining these two quantities, the length scale is found $l_k = \left( \frac{\nu^3}{\varepsilon_D} \right)^{1/4}$. The Kolmogorov entropy of the time series in this study is $S_K \left[ \frac{\text{bits}}{\text{h}} \right]$. When transforming its units by means of the Landauer principle [39] it remains $S_K \left[ \frac{\text{J}}{\text{K h}} \right]$, which is equivalent to Energy/(Temperature time), dimensionally: $\left[ \frac{\text{M}}{\text{T}} \right] \left[ \frac{\text{L}^2}{\text{t}^3} \right] \sim \left[ \frac{\text{M}}{\text{T}} \right] * \overline{\text{E}}_{\text{dissipation}}$, $\overline{\text{E}}_{\text{dissipation}}$ = average rate of energy dissipation.

Apart from the indicated dissipation energies, the atmospheric layer near the surface also experiences influences due urban densification, the change in surface roughness, heat islands, etc., which produces a connectivity picture of high complexity.

### 2.2. Entropy and Entropy Flow

In the study of transporting matter and energy between the Earth's surface and the atmosphere, mechanisms are used to understand and control vital processes such as pollutant emission and dispersion, evapotranspiration, cloud formation, wind modification or heat transfer. Taking the example of heat transfer, the entropy balance equation [40,41] is derived from Gibbs's relationship [42], from the assumption of local equilibrium, such that entropy per unit of volume, s, is written as [9]:

$$\frac{\partial s}{\partial t} = -\vec{\nabla} \cdot \vec{J}_s + \sigma \tag{1}$$

where $\sigma$ denotes the production of entropy. Or, by using the First Law of Thermodynamics, the continuity equation, the component balance equation and rewriting the local derivative of $S = V \times s$, where $V$ is the volume, the emission of entropy per unit volume ratio, s, forms the following equation [17,42]:

$$\frac{\partial s}{\partial t} = -\nabla * \left( s\vec{v} + \frac{1}{T}\vec{J}_{\text{Heat flux}} - \sum_{j=1}^{r}\frac{\mu_j}{T}\vec{J}_{\text{Diffusive mass flow, } j} \right) + \sigma, \text{ Unit S.I. } W/Km^3 \tag{2}$$

As the entropy in a system is a determinant in the order according to its extension, the evolution of a reduced atmospheric system should be related to entropy flow [42]. The idea of a reduced system suggests that, when applying a layer model, contaminants are assumed to be an external layer (of $\Delta S_E$ entropy) that embraces the inner layer of the near-ground atmosphere (of $\Delta S_I$ entropy) [17].

Linear methods can be the starting point in the study of the behavior of air pollution and meteorology data in order to carry out the analysis and prediction of temporal series. However, there are more relevant results using nonlinear tools [43,44]. Sensitive dependence on initial conditions is based on the idea that small variations in a predetermined system can produce large changes in its future [45]. This is the essence of nonlinearity, known as the butterfly effect, and even more so, chaos in its evolution [16,46,47].

A 2010–2013 study [17] of chaotic parameters showed that they can be interpreted according to thermodynamic principles of entropy, negentropy and entropic flow, $\partial S/\partial t$ [9]. A similar review of the 2017–2020 data allowed for comparison of the chaotic parameters of the meteorology-polluting interaction across both periods.

### 2.3. Kolmogorov's Entropy and Its Relationship to the Loss of Information

While chaos is well-quantified by the Lyapunov exponent, there is no universal measure of complexity [14]. Entropy can be used as an indicator of the complexity of a system (system described by the time series of this study). Connectivity is an important property of complex systems [14]; it is what stimulates the study of the Kolmogorov entropy of meteorological variables and pollutants in urbanization processes. Using the entropies, the magnitude of this connectivity is approximately quantified.

According to Farmer [48], one of the essential differences between chaotic and predictable behavior has to do with the fact that chaotic trajectories continuously generate new information while predictable trajectories do not [48,49]. In addition to providing a good definition of chaos, metric entropy provides a quantitative way to describe how chaotic a dynamic system is. In Kolmogorov's entropy [50], $S_K$ is the average loss of information [51,52] when "l" (cell side in units of information) and $\tau$ (time) become infinitesimal:

$$S_K = -\lim_{\tau \to 0}\lim_{l \to 0}\lim_{n \to \infty}\frac{l}{n\,\tau}\sum_{0....n} P_{o....n}\log P_{0....n} \tag{3}$$

It is expressed in bits of information per time and bits per iteration for a discrete system [53–55]. The order of priority in the calculation of the limits is as shown in Equation (3). First $n \to \infty$, then $l \to 0$, to remove the dependency on the chosen partition. In the preceding equation, n is the number of cells or partitions. Finally, $\tau \to 0$, which is only necessary in continuous systems. Kolmogorov's entropy difference, $\Delta S_K = S_{Kn+1} - S_{Kn}$ between cells, represents the additional information needed to know which cell ($i_{n+1}$) the system will be in the future. Therefore, this difference measures the loss of system information over time. To sum up, to calculate Kolmogorov's entropy, it is first verified that entropy is between zero and infinite ($0 < S_K < \infty$), which allows us to verify the presence of chaotic behavior. Second, the amount of information required to predict the future behavior of two interactive systems, in this case, the atmosphere and pollutants, is determined. Then,

the rate at which the system loses (or outdates) information over time is calculated. Finally, the maximum predictability horizon of the system is established, the border from which no prediction can be made, nor the elaboration of new scenarios [55]. Information loss can be calculated according to:

$$< \Delta I > \ = \ < I_{NEW} - I_{OLD} > \ = \frac{-\lambda(i_0(t))}{\log 2}. \tag{4}$$

$\lambda$ is the Lyapunov exponent, $<\Delta I>$ in [bits/h], is the loss of information. Two types of $<\Delta I>$ were calculated: one for the sum of the contribution of each P (pollutants: $PM_{10}$, $PM_{2.5}$ and CO) and another for the sum of the contribution of each MV (meteorological variables: T, WV, RH).

## 3. Materials and Methods

### 3.1. Area of Study

The city of Santiago is located at 33.5° S and 70.8° W. It contains a population of 7,508,334 inhabitants, which represents 40% of the total population of the country, on a surface of approximately 641 km$^2$. It is located in the middle of the country, at a height of about 520 m.a.s.l. The altitude above sea level increases from west to east. It is surrounded by two mountain chains: The Andes and the Coastal Mountain range. Its climate is Mediterranean. The driest and warmest months are from December to February, reaching maximum temperatures of about 35 °C in the shade (air temperature in the sun). Given its topography and the dominant meteorological conditions, there is in general a strong horizontal and vertical dispersion of pollutants generated by an important number of sources in the city (heating, vehicles, industries, etc.), especially during fall (20 March–21 June) and winter (21 June–23 September) [56]. The emissions have a tendency to increase given the also increasing population density, which implies an increase in fixed and mobile sources. In addition, the number of vehicles has increased rapidly in recent years.

### 3.2. The Data

The measures, made in the same way as in the 2010–2013 period [17], were collected from the MACAM III Network from the National Air Quality Information System [13], under Chile's Ministry of the Environment. They correspond to 3.25 years or 39 months of measurements of $PM_{10}$, $PM_{2.5}$, CO, temperature (T), relative humidity (RH) and wind speed magnitude (WV), totaling 28,463 datapoints for each variable, giving a total of 1,024,668 measurements for all communes from 2017–2020. The temporal resolution of the time series is one hour. Table 1 indicates the location, characteristics of the equipment used and the acronym of the owning institution (SINCA, in English: National Air Quality Information System).

**Table 1.** The continuous recording of the variables of this study is carried out at 2 [m] above the ground in all sensors, with the exception of the one measuring the magnitude of wind speed, which is at 10 [m].

| Station Name | Location | PM$_{10}$ | PM$_{2.5}$ | CO | T | RH | WV | OWNER |
|---|---|---|---|---|---|---|---|---|
| 1.La Florida, EML, m.a.s.l.:784 [m] | 33°30′59.7″ S 70°35′17.4″ W | Attenuation Beta-Met One 1020 | Attenuation Beta-Met One 1020 | Gas Correlation Filter IR Photometry-Thermo 48i | VAISALA HMP35A | VAISALA HMP35A | Sensor-Met One 010C | SINCA |
| 2.Las Condes, EMM, m.a.s.l.:709 [m] | 33°22′35.8″ S 70°31′23.6″ W | Attenuation Beta-Met One 1020 | Attenuation Beta-Met One 1020 | Gas Correlation Filter IR Photometry-Thermo 48i | VAISALA HMP35A | VAISALA HMP35A | Sensor-Met One 010C | SINCA |
| 3.Santiago-Parque O'Higgins, EMN, m.a.s.l.: 570 [m] | 33°27′50.5″ S 70°39′38.5″ W | Attenuation Beta-Met One 1020 | Attenuation Beta-Met One 1020 | Gas Correlation Filter IR Photometry-Thermo 48i | VAISALA HMP35A | VAISALA HMP35A | Sensor-Met One 010C | SINCA |

**Table 1.** *Cont.*

| Station Name | Location | PM$_{10}$ | PM$_{2.5}$ | CO | T | RH | WV | OWNER |
|---|---|---|---|---|---|---|---|---|
| 4.Pudahuel, EMO, m.a.s.l.:469 [m] | 33°27′06.2″ S 70°40′07.8″ W | Attenuation Beta-Met One 1020 | Attenuation Beta-Met One 1020 | Gas Correlation Filter IR Photometry-Thermo 48i | VAISALA HMP35A | VAISALA HMP35A | Sensor-Met One 010C | SINCA |
| 5.Puente Alto, EMS, m.a.s.l.:698 [m] | 33°33′01.3″ S 70°34′51.4″ W | Attenuation Beta-Met One 1020 | Attenuation Beta-Met One 1020 | Gas Correlation Filter IR Photometry-Thermo 48i | VAISALA HMP35A | VAISALA HMP35A | Sensor-Met One 010C | SINCA |
| 6.Quilicura, EMV, m.a.s.l.:485 [m] | 33°21′51.6″ S 70°44′53.9″ W | Oscillating Element Microbalance TEOM-Thermo 1400AB | Attenuation Beta-Met One 1020 | Gas Correlation Filter IR Photometry-Thermo 48i | VAISALA HMP35A | VAISALA HMP35A | Sensor-Met One 010C | SINCA |

The six monitoring stations used in this study were Florida (EML), Las Condes (EMM), Santiago-Parque O'Higgins (EMN), Pudahuel (EMO), Puente Alto, (EMS) and Quilicura (EMV), as shown in Figure 1 for their geographical distribution.

It is common that in the ordered pairs that form the sequence of data of the time series, areas of missing measurements can appear. This might be due to different factors, for example, the monitoring instrument stopped measuring due to a momentary power supply interruption [13,39,57]. These missing data were filled in through a Kriging technique [58,59]. This is a temporal and spatial geostatistical method that offers a probabilistic framework for the data analysis and its predictions based on the temporal and spatial dependency of the observations [60]. The analysis focuses on the spatial interpolation in specific time slots. The information is compared with predictive models for the different times. In addition, it is modelled according to a series of multiple times in which each spatial location is associated with a different time series [61,62]. The theory focused on the geostatistical prediction also shows the time dimension [63–65]. Hence, this technique allows for interpolation of the missing variable to be measured (polluting concentration) in a data station (defined spatial coordinates), according to similar information that is present in monitoring stations spatially close or neighboring during an analogue time interval. It is worth noticing that there is clearly no transgression of any basic statistical principles. In order to confirm this result, PM$_{10}$ time series were chosen for all communes of the 2010–2013 period with missing data problems. For each time series, 17,149 data were randomly eliminated. Using the Kriging technique, they were completed. The following statistics were obtained, see Table 2:

**Table 2.** Statistical parameters of the PM$_{10}$ time series for the periods 2010–2013 and 2017–2020 of the six communes. Measurements are shown in µg/m$^3$. Original series and in parentheses (modified series).

| Statistical Parameters | Periods | EMS | EML | EMN | EMO | EMV | EMM |
|---|---|---|---|---|---|---|---|
| Average | 2010–2013 | 62 (62) | 70 (70) | 69 (69) | 64 (65) | 79 (78) | 52 (52) |
|  | 2017–2020 | 65 (65) | 65 (65) | 69 (69) | 54 (54) | 66 (66) | 63 (63) |
| Min | 2010–2013 | 1 (1) | 1 (1) | 1 (1) | 1(1) | 0 (0) | 1 (1) |
|  | 2017–2020 | 0 (0) | 0 (0) | 0 (0) | 0 (0) | 0 (0) | 0 (0) |
| Max | 2010–2013 | 763 (763) | 686 (686) | 533 (533) | 592 (592) | 659 (659) | 770 (770) |
|  | 2017–2020 | 566 (566) | 609 (609) | 536 (536) | 807 (807) | 511 (511) | 460 (460) |
| Deviation | 2010–2013 | 42 (42) | 51 (50) | 47 (47) | 52 (52) | 57 (56) | 33 (33) |
|  | 2017–2020 | 40 (40) | 48 (47) | 46 (45) | 33 (33) | 44 (43) | 36 (36) |
| Median | 2010–2013 | 53 (53) | 59 (59) | 60 (59) | 51 (52) | 67 (66) | 46 (46) |
|  | 2017–2020 | 56 (56) | 53 (53) | 59 (59) | 47 (47) | 55 (55) | 57 (57) |
| Mode | 2010–2013 | 44 (42) | 55 (43) | 49 (37) | 40 (44) | 54 (49) | 33 (33) |
|  | 2017–2020 | 48 (48) | 42 (42) | 46 (46) | 34 (34) | 50 (50) | 48 (48) |

To validate the filling of missing data, using the Kriging technique, 17,149 data were randomly extracted from the original series of $PM_{10}$ (2010–2013). With Kriging, the extracted data were filled in. In contrast, using a cross-data validation technique, Akaike, the results are shown in Figure 2a, $r^2 = 0.80$. Figure 2b shows the totality of the data of the series (which includes missing data that have been filled in by Kriging) versus the total series of the data constructed with the formalities of the Kriging technique (a matrix of data). Since $r^2 = 0.98$, the Kriging interpolation method recreates the data matrix (Table 3).

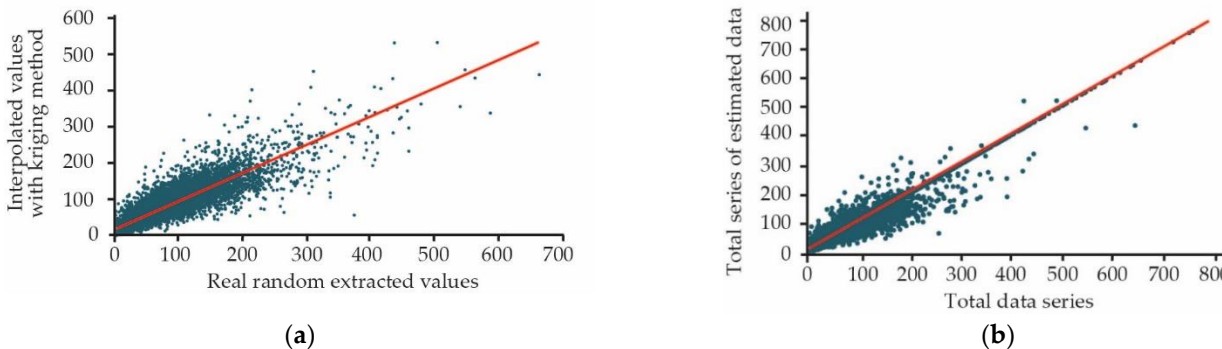

(**a**)       (**b**)

**Figure 2.** (**a**). Akaike Cross Validation; (**b**). Kriging Cross Validation.

**Table 3.** Amount of data points lost for the total measurements of each variable (28,463) and its percentage (in parentheses in the table) by commune and periods. WV: wind velocity, T: temperature, RH: relative humidity, $PM_{2.5}$: fine particulate matter, $PM_{10}$: coarse particulate matter, CO: Carbon monoxide.

| | | Stations | | | | | |
|---|---|---|---|---|---|---|---|
| **Variable** | **Periods** | **EML (1)** | **EMM (2)** | **EMN (3)** | **EMO (4)** | **EMS (5)** | **EMV (6)** |
| WV | 2010–2013 | 363 (1.3%) | 304 (1.1%) | 3065 (10.8%) | 441 (1.5%) | 2464 (8.7%) | 4799 (16.9%) |
| | 2017–2020 | 1689 (5.8%) | 2001 (7%) | 503 (1.8%) | 1392 (4.9%) | 2569 (9%) | 463 (1.6%) |
| T | 2010–2013 | 586 (2.1%) | 7123 (25%) | 648 (2.3%) | 902 (3.2%) | 736 (2.6%) | 1950 (6.9%) |
| | 2017–2020 | 6879 (24.2%) | 2076 (7.3%) | 478 (1.7%) | 1561(5.5%) | 131(0.5%) | 2463 (8.7%) |
| RH | 2010–2013 | 544 (1.9%) | 4245 (14.9%) | 597 (2.1%) | 3634 (12.8%) | 742 (2.6%) | 718 (2.5%) |
| | 2017–2020 | 1736 (6.1%) | 2061 (7.2%) | 736 (2.6%) | 351 (1.2%) | 7588 (26.7%) | 843 (3%) |
| CO | 2010–2013 | 369 (1.3%) | 118 (0.4%) | 525 (1.8%) | 842 (3%) | 205 (0.7%) | 525 (1.8%) |
| | 2017–2020 | 1526 (5.4%) | 2249 (7.9%) | 2143 (7.5%) | 1706 (6%) | 0 (0%) | 1203 (4.2%) |
| $PM_{10}$ | 2010–2013 | 340 (1.2%) | 288 (1%) | 571 (2%) | 456 (1.6%) | 257 (0.9%) | 398 (1.4%) |
| | 2017–2020 | 16 (0.1%) | 1439 (5.1%) | 664 (2.3%) | 505 (1.8%) | 600 (2.1%) | 598 (2.1%) |
| $PM_{2.5}$ | 2010–2013 | 2554 (9%) | 304 (1.1%) | 568 (2%) | 558 (2%) | 302 (1.1%) | 440 (1.5%) |
| | 2017–2020 | 1711(6%) | 61 (0.2%) | 1040 (3.7%) | 1021 (3.6%) | 978 (3.4%) | 1641 (5.8%) |

The same procedure described above was applied to the 2017–2020 period. This is represented in Figure 3a,b.

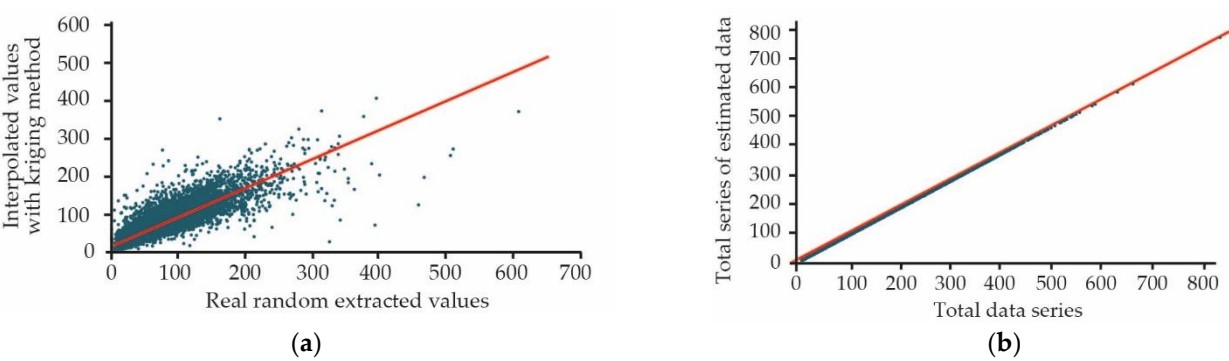

(**a**)       (**b**)

**Figure 3.** (**a**). Akaike Cross Validation, $r^2 = 0.80$; (**b**). Kriging Cross Validation, $r^2 = 0.99$.

A total of 4.785% of the data was completed with the technique described above for the two measurement periods.

### 3.3. Tools for Analysis in Nonlinear Time Series

The study of a nonlinear time series [16,39,40] starts with the establishment of two important parameters: time delay, $\tau$, and the embedding dimension, m (or $d_c$). This is carried out applying the Takens method [66]. For that purpose, the average mutual information is used [67] in the first case and the false nearest neighbors method in the second [16,68]. Other relevant information in the nonlinear analysis is the Lyapunov exponent, $\lambda$ [69], the correlation dimension, $D_C$ [14], the Hurst's Coefficient, H [14,70], the Kolmogorov entropy, $S_K$ and the correlation entropy, $K_2$. Considering that $d_0 = \|x_i - x_j\|$ is the initial distance between two arbitrary samples of the series $x_i$ and $x_j$, and if we assume that the distance $d_n = \|x_{i+n} - x_{j+n}\|$ between two samples at position n increases exponentially with n, the Lyapunov exponent is defined as $\lambda$ from the following Equation (5):

$$\lambda = \frac{1}{n} \ln \frac{d_n}{d_0} \tag{5}$$

where n is the iteration number (number of points in the time series). This means that if two points in an orbit are initially close, the exponent is calculated for a big n. If, after n iterations, the points are separated, there will be an indication of a possible chaos in such a system. A positive value of the maximum Lyapunov exponent is indicative of chaos [36,37]. For a specific time series, the sum of all the positive Lyapunov exponents defines their $S_K$ entropy and the reciprocal average time of predictability, $T_P = 1/S_K$ [16]. Effectively, the Lyapunov exponent is obtained given the equation in the boundary of big N, so that the saturation begins to be evident. For this same reason, and given the stability conditions for its measurement, there must ideally be at least 5000 datapoints available [71]. The correlation entropy, $K_2$ [14,72], is defined as:

$$K_2 = \lim_{m \to \infty} \lim_{r \to 0} \lim_{n \to \infty} \log \frac{C(m, r)}{C(m+1, r)} \tag{6}$$

where n is the number of points or data, m is the embedding dimension, and r is the radius of the circle or sphere. $K_2$ is zero, positive, or infinite for regular, chaotic, or random data, respectively. Thus, this entropy, just like Kolmogorov's, serves to establish whether a temporal series of experimental data is regular, chaotic, or random, for both pollutants and meteorological variables.

In Equation (6), C(m, r) is the sum of the correlation of the trajectory reconstructed in a time series. A method to estimate $K_2$ in experimental data is based on Grassberger and Procacia [70]. In this case, only Kolmogorov's entropy has been calculated, $S_K$, for the pollutant series and the meteorological variables, which can be seen in Table 4.

C(m, r) is the sum of the correlation for a dimension of embedding m given and is used to estimate the correlation dimension [14]. This is defined as:

$$C(r) = \frac{2}{n(n-1)} \sum_{j=1}^{n} \sum_{i=j+1}^{n} H(r - r_{ij}) = \lim_{n \to \infty} \frac{2}{n(n-1)} \sum_{i \neq j}^{n} H(r - \|x_i - x_j\|) \to C(m, r) \tag{7}$$

where $r_{ij} = \sqrt{\sum_{k=0}^{m-1} \left(x_{i-k} - x_{j-k}\right)^2}$ (the sum depends on m (embedding)), C(r) is the function of the correlation, and it can be interpreted as the number of points inside all the circles of radius r normalized to 1, when r is big enough that it includes all the points without double-counting; n is the number of data, H is the Heaviside function or (step function), ||...|| shows the norm or distance between two vectors, where Euclidean is the most used one, since it offers stronger results even during noise, and r is a real number that must be chosen carefully, since small r values make C(r) senseless, and for bigger r values C(r) does not provide valuable information.

**Table 4.** Parameters for chaos study of three pollution variables and three meteorological variables in six monitoring stations (Santiago, Chile, 2017–2020 Period).

| Parameters Station | PM$_{10}$ (μg/m$^3$) | PM$_{2.5}$ (μg/m$^3$) | CO (ppm) | Temperature (°C) | HR (%) | WV(m/s) |
|---|---|---|---|---|---|---|
| EML | | | | | | |
| $\lambda$ | 0.550 | 0.235 | 0.026 | 0.205 | 0.064 | 0.935 |
| D$_c$ | 3.451 | 1.364 | 0.580 | 2.290 | 2.029 | 3.697 |
| H | 0.922 | 0.963 | 0.933 | 0.915 | 0.942 | 0.975 |
| S$_K$ (1/h) | 0.295 | 0.596 | 0.686 | 0.355 | 0.414 | 0.515 |
| LZ | 0.234 | 0.228 | 0.018 | 0.038 | 0.087 | 0.551 |
| EMM | | | | | | |
| $\lambda$ | 0.383 | 0.614 | 0.013 | 0.184 | 0.067 | 0.937 |
| D$_c$ | 2.530 | 1.215 | 1.254 | 2.102 | 2.203 | 3.729 |
| H | 0.906 | 0.983 | 0.933 | 0.917 | 0.941 | 0.976 |
| S$_K$ (1/h) | 0.514 | 0.400 | 0.492 | 0.377 | 0.309 | 0.519 |
| LZ | 0.196 | 0.255 | 0.011 | 0.037 | 0.089 | 0.557 |
| EMN | | | | | | |
| $\lambda$ | 0.621 | 0.292 | 0.033 | 0.223 | 0.092 | 0.917 |
| D$_c$ | 2.948 | 1.276 | 2.277 | 2.280 | 2.095 | 3.735 |
| H | 0.929 | 0.960 | 0.933 | 0.916 | 0.942 | 0.973 |
| S$_K$ (1/h) | 0.242 | 0.825 | 0.412 | 0.366 | 0.308 | 0.471 |
| LZ | 0.265 | 0.233 | 0.021 | 0.042 | 0.099 | 0.539 |
| EMO | | | | | | |
| $\lambda$ | 0.550 | 0.332 | 0.046 | 0.189 | 0.081 | 0.928 |
| D$_c$ | 2.659 | 1.284 | 2.334 | 1.611 | 2.010 | 2.755 |
| H | 0.936 | 0.925 | 0.933 | 0.919 | 0.942 | 0.974 |
| S$_K$ (1/h) | 0.819 | 0.424 | 0.387 | 0.184 | 0.330 | 0.479 |
| LZ | 0.220 | 0.265 | 0.022 | 0.040 | 0.106 | 0.537 |
| EMS | | | | | | |
| $\lambda$ | 0.597 | 0.279 | 0.030 | 0.228 | 0.063 | 0.933 |
| D$_c$ | 3.535 | 1.396 | 3.302 | 2.300 | 2.306 | 3.004 |
| H | 0.921 | 0.975 | 0.933 | 0.915 | 0.942 | 0.976 |
| S$_K$ (1/h) | 0.898 | 0.422 | 0.382 | 0.357 | 0.404 | 0.489 |
| LZ | 0.204 | 0.264 | 0.018 | 0.037 | 0.071 | 0.556 |
| EMV | | | | | | |
| $\lambda$ | 0.516 | 0.304 | 0.031 | 0.170 | 0.065 | 0.915 |
| D$_c$ | 1.148 | 1.419 | 2.149 | 1.577 | 1.947 | 2.355 |
| H | 0.931 | 0.966 | 0.933 | 0.919 | 0.942 | 0.975 |
| S$_K$ (1/h) | 0.267 | 0.463 | 0.490 | 0.171 | 0.428 | 0.395 |
| LZ | 0.231 | 0.296 | 0.019 | 0.029 | 0.085 | 0.544 |

Equation (7) can also be written as follows:

$$C(r) = \lim_{n \to 0} \frac{1}{n^2} \left[ \text{number of pairs } (x_i, x_j) \text{ such that } |x_i - x_j| < r \right] \tag{8}$$

$C(r)$ is calculated by varying r from 0 towards the highest possible number of $\|x_i - x_j\|$. For values of r which are sufficiently small and for a big quantity of data, $C(r)$ behaves following the power law of the kind:

$$C(r) \sim r^{D_C} \tag{9}$$

$D_C$ is the correlation dimension (or correlation function). Given Equation (9), and taking logarithms on each side, $D_C$ is obtained through a log $C(r)$ vs. log r graph. If $D_C$ is not saturated in any value while m increases, then the process is random (or stochastic). On the contrary, if $D_C$ is saturated in some value, then the time series is determinist [73]. The correlation dimension, $D_C > 5$ essentially implies random data.

Lyapunov's exponent, $\lambda$, characterizes the nature of the temporal evolution of close trajectories in phase space and is considered a key component of chaotic behavior. Thus, it can be stated [72] that the correlation entropy, $K_2$, is a lower bound of Kolmogorov's entropy, $S_K$. That is,

$$K_2 \sim S_K \tag{10}$$

These relationships are part of the numerical calculation procedure through a software that is applied to each time series (both pollutants and meteorological variables), each of 28,463 data, once they do not present missing data. The nonlinear analysis of the temporal series according to chaos theory [14,17,73] also included the Iterated Function Systems fragmentation test (IFS) as a method of data analysis. The use of symbolic dynamics [74] allows us to calculate the Lempel–Ziv complexity (LZ) relative to white noise. When the data are chaotic, these LZ are distributed, forming localized groups on the surface. The chaotic series conditions are met for the two studied periods, since $0 < LZ < 1$, for the data, as shown in the calculation for all the communes.

Table 4 shows that all the time series of the study variables are chaotic.

## 4. Results

One of the objectives of the study was to examine the behavior (determinist, chaotic or random) of the three time series of pollutants ($PM_{10}$, $PM_{2.5}$ and CO) and three time series of meteorological values (magnitude of wind speed, relative humidity, and temperature). The most important indexes (or metrical quantities) in the analysis of nonlinear time series are: lagging time, $\tau$; embedding dimension, m; maximum positive Lyapunov exponent, $\lambda$; correlation dimension, $D_C$; Kolmogorov's entropy $S_K$; and Hurst exponent, H, which allow to quantitatively demonstrate if a system represented, in this case through time series, is chaotic. Thus, if the correlation dimension is under 5.0, then the Lyapunov exponent is positive, Kolmogorov's entropy has a finite and positive value, and chaos would be present in the time series of the study. Kolmogorov's entropy $S_K$ is a feature of the degree of chaos for a nonlinear system. This value is proportional to the rate of loss of information in the system.

Table 4 provides the first important result: the verification of the chaoticity of all time series for the two study periods. For the six time series of hourly measurements, $D_C$ was saturated at the values presented in Table 4. The correlation dimension obtained for each series is $D_C < 5$.

On the other hand, a different approach to the problem of relating the measurement locations with the changes in soil properties is shown through graphic representations, specifically densification and changes in roughness by high-rise building. Finally, it is possible to observe the effect of these changes in the meteorological variables [75] of temperature, magnitude of wind speed, and relative humidity in the measured locations.

Figure 4 shows the quotient between the sum of the Kolmogorov entropies of the meteorological variables and that of the pollutants, for the two periods 2010–2013 [17] (Series 1) and 2017–2020 (Series 2), in the same communes of the metropolitan region. The $C_K$ calculation, using the $S_K$ data from Table 4 by commune, is carried out with the formula:

$$C_{K_{COMMUNE}} = \frac{\sum \text{Entropies of meteorological variables}_{COMMUNE}}{\sum \text{Entropies of pollutants}_{COMMUNE}} \tag{11}$$

As indicated in Table 4 and Figure 4, the entropy of the considered pollutants increased compared to that of the meteorological variables (T, WV, RH), for the 2017–2020 period. Because of that, the pollutant system has become more complex [14]. There was a decline in the $C_K$ index according to the communes where the monitoring stations are located, an antecedent that would appear insubstantial if it were not for Figure 5, below. This figure establishes a relationship between the $C_K$ index and the constructed area, preferably of height (explained in the Urban Densification Introduction section referring to the communes of the study):

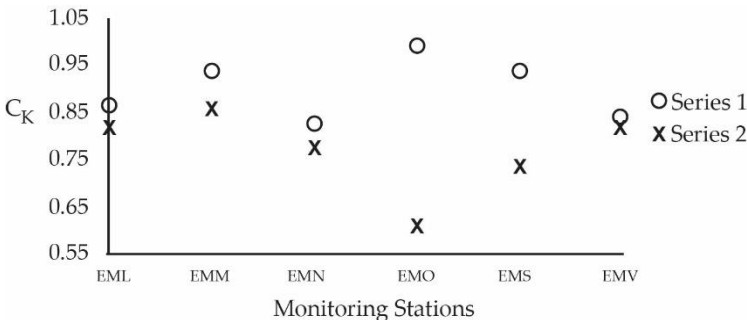

**Figure 4.** Plotted $C_K$ ratio according to monitoring station (Figure 1) for the considered periods: 2010–2013 (Series 1) [17], 2017–2020 (Series 2).

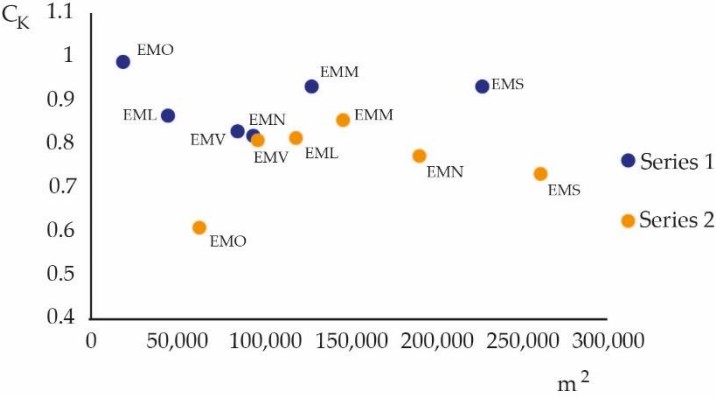

**Figure 5.** $C_K$ versus m² residential data built in the communes of the study [MVU, INE,2020]. Series 1: 2010–2013, Series 2: 2017–2020 (period of urban densification).

The decays in $C_K$ are similar to those in Figure 4. This is a sign that the constructed index (using Kolmogorov entropy) can provide information about the effect of massive and high-rise construction on pollutants and meteorological variables.

The percentage variation of $C_K$ (22%) is high in the EMS pre-Andean commune, since it contains the largest population of all the communes of Chile and experiences a very strong urbanization process. Of the 86.74 km² of the commune, 31.38 km² (36.18%) correspond to the territory occupied by the current urban sites. The EMO commune (38%) is highly complex because apart from experiencing a strong housing construction process, it is a traffic area for three main roads. This commune contains the largest airport in Chile (great movement of people and products), with a large industrial park. EMN (7%) is close to a large metropolitan park (Cerro San Cristobal, 880 masl) and a river circulates on one part of the periphery of EMN (Rio Mapocho, flow 6.3 m³/s). EMM (7%) is located in a transition zone between the capital's valley and the foothills of the Andes and contains the Aguas de Ramón Park. EML (6%) is divided into four zones, and the largest zone includes the San Ramon geological fault (31,273 km²). Much of the fault area is mountainous and uninhabited, with great risk of alluvium. EMV (3%) is a peripheral commune, far from the center of Santiago. The topography of the land is mostly flat and at times undulating by certain low hills that go unnoticed. Being surrounded by several hills, such as Cerro Renca, it gives a feeling of being in an independent basin. The basalt-type terrain favors a good dispersion of gases and pollutants.

Additionally, using the Hurst coefficient and calculating the fractal dimension, the greater chaoticity of the period 2017–2020 compared to 2010–2013 is demonstrated. Calculating the quotient between the averaged fractal dimension of meteorological variables and the averaged fractal dimension of pollutants, by commune and period, shows an increase in chaoticity, Figure 6 shows that the most square meters were built in the period 2017–2020 (see Appendix A for Table and $C_D$ definition):

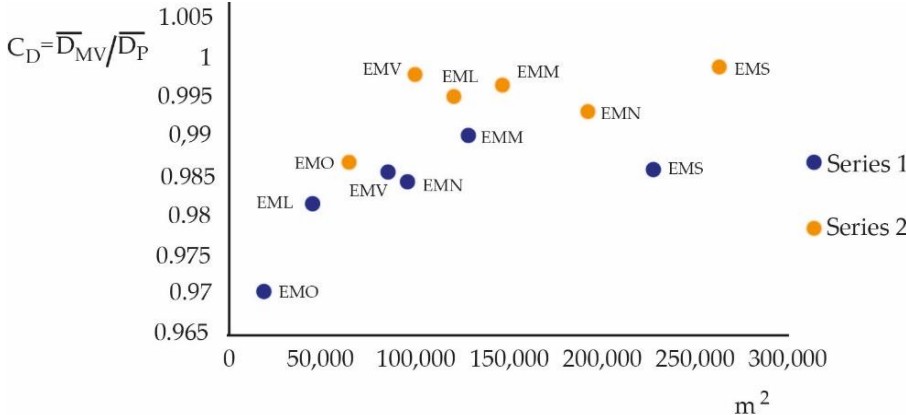

**Figure 6.** It indicates that series 1 of the period 2010–2013 corresponds to a less chaotic regime than time series 2 of the period 2017–2020, confirming the $C_K$ result.

In other words, the fractal dimension associated with meteorological variables and pollutants is greater in the period 2017–2020. This attribute is related to the degree of roughness that the time series presents and allows for mathematically describing objects that present a high degree of complexity, self-similarity or chaos. The data that make up the time series are measured at ground level by instruments located at the same points at different periods (2010–2013, 2017–2020, 7 years). Other factors exist that may contribute to series roughness. Chile has updated standards of pollutant concentration levels (PM$_{10}$, PM$_{2.5}$, CO, etc.) that allow it to face pre-emergency and emergency situations. Restrictions on vehicular circulation and the use of chimneys (for example), have been used for more than 15 years, mainly in winter (which suggests some control over the concentration levels of pollutant). Additionally, the central zone of Chile is experiencing a drought that has lasted for more than 20 years (and has been getting worse), affecting localized micrometeorology. However, there is enough literature, already mentioned, that explains the effect of intensive urbanization on micrometeorology even in the presence of broader climate change effects. All this has an impact on the roughness of the series. However, it does not fully explain the variation in $C_K$ or $C_D$. The roughness of the series is also affected by the change in the roughness of the soils due to urbanization processes that leads to urban densification (from cases of densification in contexts that favor certain mitigation to other contexts of little or no influence (cruder)). Urbanization is in closer contact with localized micrometeorology (later, Figures 7 and 8). This shows that what $C_K$ indicates is confirmed by $C_D$.

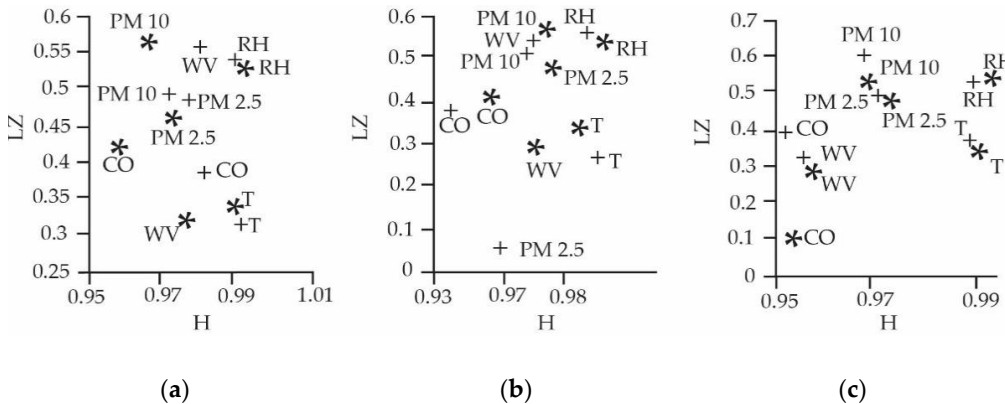

**(a)**         **(b)**         **(c)**

**Figure 7.** LZ v/s H, 2010–2013 period (Table II, [17]), symbol in graphs, according to each monitoring station: (**a**) EML (*) y EMM (+), (**b**) EMN (*) y EMO (+), (**c**) EMS (*) y EMV (+).

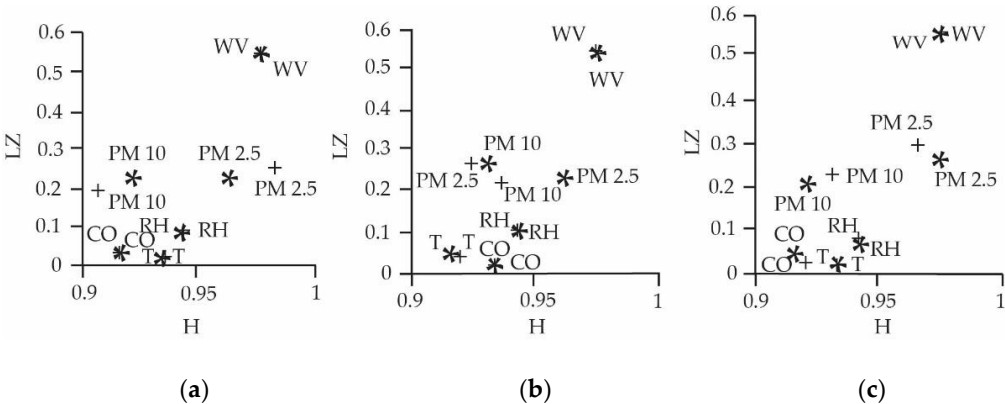

**Figure 8.** LZ v/s H, 2017–2020 period (Table 4), symbol in graphs, according to each monitoring station: (**a**) EML (*) y EMM (+), (**b**) EMN (*) y EMO (+), (**c**) EMS (*) y EMV (+).

Table 5 reports on the loss of information (bits/h), which was faster in the 2017–2020 period, and was bulkier and rougher, corroborating Figures 4 and 5. This is characteristic of chaotic systems [76].

**Table 5.** <ΔI> [bits/hr] = loss information as the sum of the contribution of each P (pollutants: $PM_{10}$, $PM_{2.5}$, CO) and sum of contribution of each MV (meteorological variables: T, WV, RH).

| Stations | EML | EMM | EMN | EMO | EMS | EMV |
|---|---|---|---|---|---|---|
| Periods | <ΔI>$_P$; <ΔI>$_{MV}$ | <ΔI>$_P$; <ΔI>$_{MV}$ | <ΔI>$_P$; <ΔI>$_{MV}$ | <ΔI>$_P$; <ΔI>$_{MV}$ | <ΔI>$_P$; <ΔI>$_{MV}$ | <ΔI>$_P$; <ΔI>$_{MV}$ |
| 2010–2013 | −5.341; −6.079 | −5.039; −6.859 | −4.537; −6.357 | −3.271; −6.633 | −4.656; −6.955 | −3.825; −6.899 |
| 2017–2020 | −2.694; −4.000 | −3.355; −3.957 | −3.142; −4.092 | −3.083; −3.980 | −3.010; −4.066 | −2.827; −3.820 |

According to the data, the 2017–2020 period had greater urban densification, yet the loss of information was faster than in the 2010–2013 period [17].

When observing the behavior, according to the LZ and H in Figures 7 and 8, we can see what the graphs indicate for the meteorological variables and pollutants considered for the two periods and locations of this study:

(i) A delay in temperature and relative humidity for the 2017–2020 period compared to the 2010–2013 period. Chile is experiencing a prolonged drought period, and that possibly has high significance in an area built in a geographical basin.

(ii) A greater presence of wind in the period 2017–2020 in the layers adjacent to the ground. This effect is compatible with a change in the roughness of the surface for the 2017–2020 period compared to the 2010–2013 period. This is illustrated in Table 6.

**Table 6.** Comparative table of the amount of wind speed according to persistence, H, and LZ complexity, where > means increase, and = means the same value.

| Stations | EML | EMM | EMV | EMN | EMS | EMO |
|---|---|---|---|---|---|---|
| Periods | H; LZ | H; LZ | H; LZ | H; LZ | H; LZ | H; LZ |
| 2010–2013 | 0.976; 0.320 | 0.980; 0.558 | 0.956; 0.325 | 0.968; 0.286 | 0.957; 0.293 | 0.968; 0.538 |
| 2017–2020 | 0.975 (=); 0.551 (>) | 0.976 (=); 0.557 (=) | 0.975 (>); 0.544 (>) | 0.973 (>);0.539 (>) | 0.976 (>); 0.556 (>) | 0.974 (>); 0.537 (=) |

In four of the six communes, there is an increase in LZ and persistence (H) for the wind, which indicates an increase in its turbulence (Table 6).

(iii) There is also an advance of $PM_{2.5}$, fine particulate, which has more serious consequences for people's health, unlike $PM_{10}$.

(iv) The chaotic treatment of time series delivers the following parameters: λ; Dc; H; $S_K$; and LZ, which are unique for each time series of the study period. Considering that each monitoring station that records the data, see Figure 9, is in the nearest neighbor condition, some similarity in values of complexity and persistence is to be expected. For

the 2017–2020 period, they point to the same conclusion, a decline in these parameters for relative humidity compared to 2010–2013, which is also compatible with a significant decrease in winter rains.

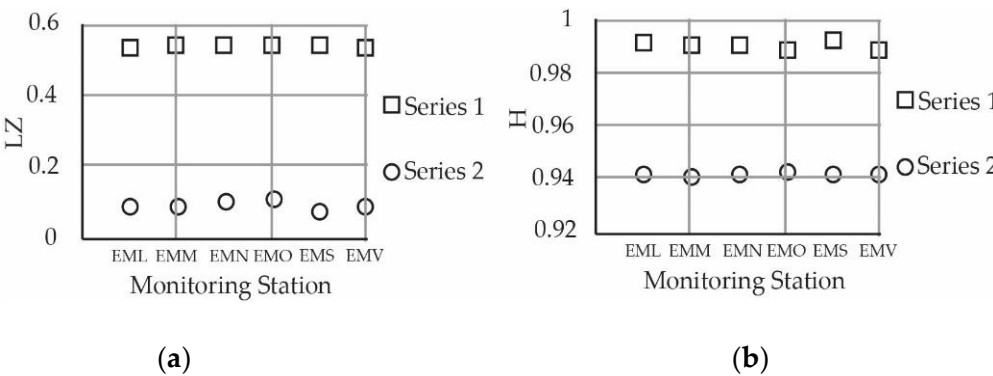

(**a**)　　　　　　　　　　　　　　　　　　　　　　　　(**b**)

**Figure 9.** Series 1 represents the 2010–2013 period and series 2 represents the 2017–2020 period for LZ and H in (**a**,**b**) for relative humidity.

(v) The periods of the year in which the temperature oscillates between more extreme values have increased. The central region of Chile was of Mediterranean climate of marked seasons: autumn, winter, spring, and summer [77]. At present, those nuances have been lost, and we find two seasons: cold and dry winter and very warm in summer; this marked loss of seasons influences measurements. The time series for the 2010–2013 period tell the story of a stage of the rainiest urban climate, marked by quite extreme events (landslides due to rains, floods, etc.) that have not been seen for many years. Figure 9 presents LZ and H of temperature in the measurement periods.

Referencing Figure 10, the monitoring equipment measured the hourly temperature for the period 2010–2013 (Series 1), capturing a temperature dominated by an urbanized environment, but which was comparatively lower than the period 2017–2020 (Series 2), in which the average temperature, by station, is larger but more volatile. The average temperature by periods, $\overline{T}_{2010–2013} = 15.64\,°C$ and $\overline{T}_{2017–2020} = 16.20\,°C$, shows an increase of 0.56 °C.

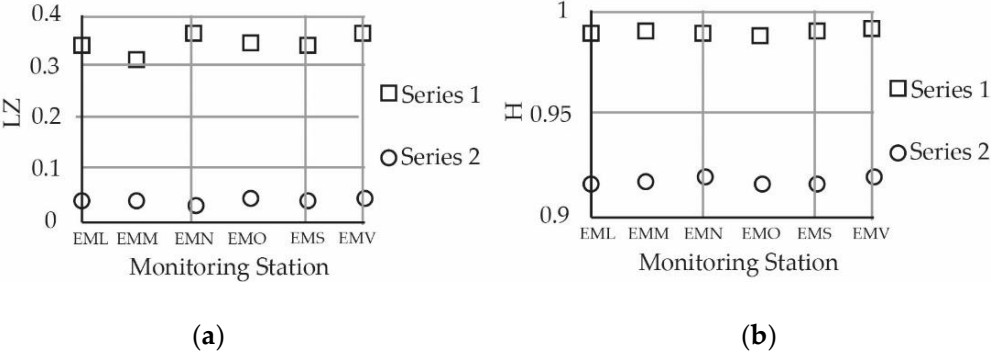

(**a**)　　　　　　　　　　　　　　　　　　　　　　　　(**b**)

**Figure 10.** Series 1 represents the 2010–2013 period and series 2 represents the 2017–2020 period for LZ and H in (**a**,**b**) for the temperature.

## 5. Discussion

The total area of this study corresponds, approximately, to 535.4 km$^2$ (assuming, roughly, a square would have sides of 23.14 km) distributed in six nonperipheral communes of the metropolitan region of Santiago de Chile. The water shortage has been investigated in Chile [78–80] as it has worsened. The drought affects central and northern zones of the country. A geographic basin exists in the central zone, where Santiago sits, and this has caused historical ventilation problems [56].

There are two meteorological phenomena that hinder an adequate dispersion of the pollutants in this city. These are thermal inversion of subsidence and a radiative one [56]. It has been possible to establish that the region of Santiago has two types of synoptic conditions directly linked with the episodes of high atmospheric pollution, which are called the A type (Anticyclonic, A in Spanish) and the BPF type (low prefrontal, BPF in Spanish) [56].

Quantitative studies of urbanization processes and their relationship with urban densification, change in roughness, as well as with pollutant and meteorological measurement techniques, have been published [9,11,12,75,79,81]. This work differs in that it uses the formalism of chaos theory [21,35], that allows for establishing a relationship between urbanization processes and the temporal evolution of the concentration of pollutants and meteorological variables [17].

This qualitative study uses the Lyapunov's exponent, which is a chaotic parameter, to quantify the loss of information. This loss was faster in the 2017–2020 period (Table 5), with greater urban densification compared to 2010–2013. This would indicate a natural connection to a more chaotic system, relative to intensive changes in urban density and roughness (Period 2017–2020). The Kolmogorov entropy [21] of the measured variables of meteorology and pollutants gives the entropic fluxes. There is also the effect of Kolmogorov cascade, which contributes to the process of heat dissipation in the atmospheric layers next to the ground, intensifying the energy transfer in turbulence on a large scale [25]. The indexes constructed, in a first approximation, $C_K$ and $C_D$ together with $<\Delta I>$, and LZ and H show variations directly related to the increase in urban densification and surface roughness (square meters of high-rise buildings) (Figures 5–8).

Chaos theory shows that meteorological data are also disturbed in their comparison between periods (2010–2013 and 2017–2020) by the effect of the entropy flow of pollutants, urban densification, and also by the effect of the Kolmogorov cascade. The wind time series for the period 2017–2020 shows that through the Lempel–Ziv algorithm it is possible to find new data subsequences inside it, which increases the complexity counter. The development of urban canyons generated by high-rise buildings offers the wind a more intricate mobility scenario. This produces a more chaotic, more complex series when compared with the period 2010–2013 (Table 6). It happens the other way around with relative humidity (RH) and temperature (T) (Figures 9 and 10). This loss of complexity in the RH and T variables can be related, for the period 2017–2020, with less new information in the data chains of their time series, and fewer variations in RH and T between autumn, winter, spring and summer, as shown in Figures 9 and 10. As indicated, the central zone of Chile currently offers a dry and cold winter and a very hot summer. These current climate dynamics, which in the last 30 years have led to a more limited seasonality, are to be considered along with process of increasing urbanization.

The wind plays a significant role in the transport and dispersion of pollutants, in addition to being influenced by local conditions of relief and roughness [12,56]. When its speed increases, the greater is the volume of air that moves per unit of time through the area where a source of pollutant emission is located. This turbulent surface layer is where pollutants are transported and dispersed. Table A1 compares the LZ and H values for the wind of both periods and presents the increase in complexity in 2017–2020.

Temperature participates in the daily dynamics of warming–cooling of the Earth's surface, with a great effect on the location of the base of the subsidence inversion layer and the intensity with which it affects Santiago de Chile. This complex structure of urban atmosphere also supports the effect of thermal islands [3,7,9,56].

Other studies that analyze the data on the relationship between RH and urban growth in the areas around the measuring instruments also show a decreasing trend in RH (Figure 9a,b). In the case of this study, there is a decline in RH between the periods 2010–2013 of 60%, and 2017–2020 of 56%.

The results from this study, along with those of several others, press for a more rigorous urban planning, the use of new construction materials and mitigation measures [82–88] both in time and space.

## 6. Conclusions

Through a qualitative study of time series that constitute the database of two time periods, it is shown that they satisfy the criteria to be defined as chaotic (Dc < 5, $\lambda$ > 0, 0.5 < H < 1, 0 < LZ < 1, $S_K$ > 0). In general, chaos theory studies dissipative, complex, or connected (nonlinear) phenomena, which are also of low predictability, entropic and irreversible. This allows us to anticipate the nature of the processes that have been induced by human activity at the study site. There is a high probability that this could also be shown in many other parts of the world.

The $C_K$ quotient for the 2017–2020 period with higher urban densification are lower for the six communes in Santiago when compared with the values of the 2010–2013 period (Figure 4) of lower densification. Therefore, it is proven that for the 2017–2020 period, the value of the persistence of the meteorological variables of temperature and relative humidity is lower than the value calculated for the $PM_{10}$ and $PM_{2.5}$ pollutants, respectively. This can be interpreted, in a first approach, as a decay in the initial conditions of the natural micrometeorology. The meteorological variables (T, RH, WV) are of basic use for any climatological model. Urban densification contributes to the prominence of pollutants according to the comparative demonstration in the 2017–2020 and 2010–2013 periods.

The Hurst exponent and the fractal dimension are related to the roughness of the time series. The fractal dimension is a statistical quantity that allows us to mathematically describe objects that present a high degree of complexity. The interperiod comparison of the time series of meteorological variables and pollutant concentration shows an increasing change in complexity in favor of the period 2017–2020, a period of urban growth.

**Author Contributions:** Conceptualization, P.P. and E.M.; methodology, P.P. and E.M.; software, P.P. and G.S.; validation, P.P. and E.M.; formal analysis, P.P.; investigation, P.P.; resources, E.M.; data curation, E.M.; writing—original draft preparation, G.S. and P.P.; writing—review and editing, G.S., P.P. and E.M.; visualization, E.M.; supervision, P.P.; project administration, P.P.; funding acquisition, P.P. All authors have read and agreed to the published version of the manuscript.

**Funding:** This research received no external funding.

**Institutional Review Board Statement:** Not applicable.

**Informed Consent Statement:** Informed consent was obtained from all subjects involved in the study.

**Data Availability Statement:** The data were obtained from the public network for online monitoring of air pollutant concentration and meteorological variables. The network is distributed throughout all of Chile, without access restrictions. It is the responsibility of SINCA, the National Air Quality Information System, dependent on the Environment Ministry of Chile. The data for the two study periods will be available for free use on the WEB page: URL: https://sinca.mma.gob.cl. Accessed on 4 April 2021.

**Acknowledgments:** The authors gratefully acknowledge the Direction for Research that funded this study through Project LPR20-02 and the Department of Physics, both part of Universidad Tecnológica Metropolitana. The authors would also like to thank the support from Dirección de Investigación e Innovación through DIREG Project number 04/2019 and Applied Mathematical and Physics Department at Faculty of Engineering, Universidad Católica de la Santíisima Concepción.

**Conflicts of Interest:** The authors declare no conflict of interest.

## Appendix A

It is also possible to analyze the time series from another point of view, using the Hurst Exponent, H. Through the fractal dimension, D = 2 − H, the roughness of the time series is analyzed, allowing us to describe objects that present a high degree of complexity, comparing them according to the study periods.

If 0.5 < H < 1, these are time series that present persistence (long-term memory effects). Theoretically, what happens in the present will affect the future forever, and all current

changes are correlated with all future changes. In the case of the present study, it means that the time series are persistent, since 0.5 < Hurst exponent < 1.0.

**Table A1.** All the values of the Hurst exponent (H) for all the variables of interest by studied commune and the two periods (2010–2013 and 2017–2020).

| | | $PM_{10}$ | $PM_{2.5}$ | CO | T | HR | WV |
|---|---|---|---|---|---|---|---|
| EML | 2010–2013 | | | | | | |
| | H | 0.967 | 0.973 | 0.959 | 0.989 | 0.991 | 0.976 |
| | D | 1.033 | 1.027 | 1.041 | 1.011 | 1.009 | 1.024 |
| | 2017–2020 | | | | | | |
| | H | 0.922 | 0.963 | 0.933 | 0.915 | 0.942 | 0.975 |
| | D | 1.078 | 1.037 | 1.067 | 1.085 | 1.058 | 1.025 |
| EMM | 2010–2013 | | | | | | |
| | H | 0.972 | 0.977 | 0.981 | 0.991 | 0.990 | 0.980 |
| | D | 1.028 | 1.023 | 1.019 | 1.009 | 1.010 | 1.02 |
| | 2017–2020 | | | | | | |
| | H | 0.906 | 0.983 | 0.933 | 0.917 | 0.941 | 0.976 |
| | D | 1.094 | 1.017 | 1.067 | 1.083 | 1.059 | 1.024 |
| EMN | 2010–2013 | | | | | | |
| | H | 0.972 | 0.974 | 0.953 | 0.989 | 0.991 | 0.968 |
| | D | 1.028 | 1.026 | 1.047 | 1.011 | 1.009 | 1.032 |
| | 2017–2020 | | | | | | |
| | H | 0.929 | 0.960 | 0.933 | 0.916 | 0.942 | 0.973 |
| | D | 1.071 | 1.04 | 1.067 | 1.084 | 1.058 | 1.027 |
| EMO | 2010–2013 | | | | | | |
| | H | 0.965 | 0.955 | 0.937 | 0.992 | 0.989 | 0.968 |
| | D | 1.035 | 1.045 | 1.063 | 1.008 | 1.011 | 1.032 |
| | 2017–2020 | | | | | | |
| | H | 0.936 | 0.925 | 0.933 | 0.919 | 0.942 | 0.974 |
| | D | 1.064 | 1.075 | 1.067 | 1.081 | 1.058 | 1.026 |
| EMS | 2010–2013 | | | | | | |
| | H | 0.969 | 0.973 | 0.953 | 0.990 | 0.992 | 0.957 |
| | D | 1.031 | 1.027 | 1.047 | 1.010 | 1.008 | 1.043 |
| | 2017–2020 | | | | | | |
| | H | 0.921 | 0.975 | 0.933 | 0.915 | 0.942 | 0.976 |
| | D | 1.079 | 1.025 | 1.067 | 1.085 | 1.058 | 1.024 |
| EMV | 2010–2013 | | | | | | |
| | H | 0.967 | 0.970 | 0.952 | 0.989 | 0.989 | 0.956 |
| | D | 1.033 | 1.03 | 1.048 | 1.011 | 1.011 | 1.044 |
| | 2017–2020 | | | | | | |
| | H | 0.931 | 0.966 | 0.933 | 0.919 | 0.942 | 0.975 |
| | D | 1.069 | 1.034 | 1.067 | 1.081 | 1.058 | 1.025 |

The fractal dimension (D) of the meteorological variables and of the pollutants is greater in the period 2017–2020. If the average value of the fractal dimension, D, is calculated for the pollutant time series, (P), and for the meteorological variables, (MV), then a quotient is constructed, $C_D = \overline{D}_{MV}/\overline{D}_P$ (similar to $C_K$):

$$C_{D,COMMUNE} = \frac{\sum \text{fractal dimensions of the meteorological vaiables}_{COMMUNE}}{\sum \text{fractal dimensions of the pollutants}_{COMMUNE}} \tag{A1}$$

and it is plotted considering the square meters of built surface associated with each commune and according to the two periods of the study, thus Figure 4 is obtained.

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
