# Peer review of "Urban Densification Effect on Micrometeorology in Santiago, Chile: A Comparative Study Based on Chaos Theory"

_sustainability, doi:10.3390/su14052845_

Round 1

Reviewer 1 Report

Interesting work is presented with very few comments to highlight, among which the following are mentioned:

  • It is suggested to review the presentation format of the equations in line 122.
  • Check the required text format and the representation of all the equations shown, since equation (5) stands out among the others.
  • General review of grammar, and formatting of multiple references.
  • Capital letters at the beginning of sentences (lines 77, 576).
  • Change the structure from (PM2.5 particulate matter, PM10, and Carbon Monoxide) to (Carbon Monoxide, PM2.5, and PM10 particulate matter).
  • Reference figure 1.
  • Improve the quality in figure 5, since quality is lost in the indicators of the Y-axis.
  • The contribution in terms of the state of the art is not clear.

Reviewer 2 Report

Dear Authors, 

I thoroughly enjoyed reading this work. The concept of trying to understand if air pollution and urban microclimate are chaotic is very interesting. I would have liked to see other pollutants instead of CO, as that one is very source dependant. NO2 or Ozone would have been more interesting. Maybe for a future work. please take specific note of problem number 4 below. 

A few comments and problems that I found:

  1. Line 296: the chapter header should 3 and not 2.
  2. Line 350-351 and line 358-359: these two sentences are a repeat of one another. Oddly enough, the numbers are different. 17149 and 17150. Please use the correct one and remove one of those instances.
  3. Line 369: should be table 3 and not tabla 3 (in Spanish?)
  4. Line 376: please perform the kriging analysis to the 2017-2020 period as well. a few years have passed, and just because it worked well to the first period, doesn't guaranty it will work well to the second period.
  5. Line 436: from my understanding, this paragraph is the starting point for the results. You report results and stop discussing methods. Please move to results, or change to methods.
  6. Line 662: “RH y T” I believe it should be and (&) and not Y. I’m assuming Spanish got in again.
  7. Some chapter and subchapter headers have numbers, and some do not. Please make sure to be consistent.

Round 2

Reviewer 1 Report

The suggestions have been attended. I do not have any more suggestions.

This manuscript is a resubmission of an earlier submission. The following is a list of the peer review reports and author responses from that submission.

Round 1

Reviewer 1 Report

This manuscript describes the use of a series of well known chaos and data roughness models to show the effect of urban development on databases of airborne pollutants and meteorological variables over compared timeframes. The authors show that urban densification affects micrometeorology and dispersion of the pollutants studied. The following suggested edits should be considered:

Line 14: revise "The concentration of athropocentric... ." to "The concentration distribution of anthropocentric... ."

Lines 20-22: revise capitalization as follows: Lyapunov exponent, correlation dimension, Hurst coefficient, correlation entropy, Lempel-Ziv complexity, and fractal dimension.

Line 37: replace "... and are characterized... ." with "... which are characterized... ."

Line  45: delete the phrase "in its inside"

Line 47: replace "had" with "has" and "translated" with "translates"

Lines 51-52: the following phrase seems like a fragment at the end of the sentence ", the training zone, the superficial layer, and the stable boundary layer" ; may have meant this to be explanatory; if so, it could be enclosed in parentheses "... during the night (the training zone, the superficial layer, and the stable boundary layer) [3]."

Line 53: add word "... wind (turbulence) being affected by... ."

Line 62: add words "...to 2-5 times the average... ."

Line 65: replace the word "expansion" with "distribution"

Lines 121-122: revise capitalization as follows: "... Hurst's coefficient, (0.5<H,1), correlation dimension (Dc<5), correlation entropy (SK >0), and complexity... ."

Line 168: add subscript "hmax = 2.5havg"

Line 171: replace period with question mark "... literature?"

Line 190: verify author name for reference 28; "von Weizsacker" or "Weizsacker" ?

Line 204: remove comma after "pollutants"

Line 230: replace "from" with "forms"

Line 233: add word "... the above equation follows:"

Line 298: suggest a semicolon "... in 1990; in 2020... ."

Line 304: replace word "was" with "were"

Line 310: add commas "... buildings, and thus the verticalization of the city,... ."

Line 347: verify number "4,6" 

Line 350: add word "... square meter areas built... ."

Line 355: remove "_" at end of sentence

Line 356: remove new paragraph indentation

Line 364: replace "... the alteration of... ." with "... an alteration of... ."

Line 365: add word "... is caused by the... ."

Lines 383-384: The following appears to be an incomplete sentence: "Quantify, as a first approximation, the magnitude of this connectivity."

Line 415: replace word "Pollutions" with "Pollutants"

Line 417: replace "Table 2" with "A summary of <ΔI>(see Table 2 for station definitions)... ."

Line 422: suggest adding commas to "7,508,334"

Line 428: replace word "bad" with "strong"

Line 442: replace "... , the radiative... ." with "... , radiative... ."

Lines 446-447: define the acronyms "A" and "BPF"

Line 450: add word "The measures, made in the same... ."

Line 452: replace "...the ministry of the Environment Ministry of Chile." with "... Chile's Ministry of the Environment."

Line 455: add word "... 1,024,668 measurements for all... ."

Line 463: replace "Figure 2" with "Figure 1"

Line 469: replace "... the second missing element appears." with "... areas of missing measurements can appear."

Line 471: capitalize "Kriging"

Line 478: replace "... interpolating... ." with "... for interpolation of... ."

Line 485: capitalize "Kriging"

Line 497: add word "... of data points lost... ."

Line 503: replace "... completed with described technique,... ." with "... completed with the technique described above,... ."

Line 524: replace "... 5000 data available." with "... 5,000 data points available."

Line 532: remove words "also for"

Line 547: add word "... since small r values make... ."

Line 551: replace "numero de pares" with " number of pairs" and "tales que" with "such that"

Line 574: add words "... under 5.0, then the Lyapunov... ."

Line 575: replace "... is positive and Kolmogorov's entropy has a finite and positive value, then the... ." with "... is positive, Kolmogorov's entropy has a finite and positive value, and the... ."

Line 592: add words "these LZ values are distributed... ."

Line 593-594: add words "The chaotic series conditions are met..." and replace "for the chaotic data, which is shown... ." with "for the data, as shown... ."

Line 600: replace "pollutions" with "pollutants"

Line 611: replace "... important result, the ..." with "... important result; the..." and "... all-time..." with "... all time..."

Line 651: replace "... in the largest zone is the... ." with "... and the largest zone includes... ."

Line 652: add word "... Much of the fault area is... ."

Line 659-660: add word "... Calculating the quotient... ."

Line 663: comment - Figure 4 does not appear to be referenced in the manuscript text.

Line 672: suggest deleting the initial phrase "On the other hand" and beginning a new paragraph with something like "Other factors exist that may contribute to series roughness. Chile has updated... ."

Line 677: replace "Chile experiences a drought that lasts for..." with "Chile is experiencing a drought that has lasted for..." and add a parenthesis after the word "worse".

Line 679-680: suggest replacing "... even in the processes of climate change." with "... even in the presence of broader climate change effects."

Line 692: add words "...Hurst coefficient in Figures 5 and 6, we... ."

Line 705: correct year range to "2010 - 2013"

Line 713: report 2010 - 2013 LZ values to three significant digits

Line 716: correct "PM2.5" and replace word "with" with "which has"

Line 718: replace "...to the finest material." with "... for the finest material to infiltrate."

Lines 722-723: change from "Figure 1" to "Figure 7" and replace "... some similarity in some values... ." to "... some similarity in values of... ."

Line 745: suggest beginning sentence with "Referencing Figure 8, the monitoring... ."

Line 748: begin sentence with "The average temperature... ."

Line 749: add word "..., shows an increase of ... ."

Lines 753: replace "is" with "has been"

Line 754: suggest replacing "... the central zone and the north of... ." with "central and northern zones of... ."

Line 755: suggest replacing "In the central zone is located the geographic basin... ." with "A geographic basin exists in the central zone... ."

Line 756: replace "... which cause... ." with "this has caused... ."

Line 758: replace "... in roughness and with... ." with "... in roughness, as well as with... ."

Line 764: replace "Which" with "This loss" and "Tabla 5" with "Table 5"

Line 765: replace "Which" with "This"

Line 766: replace "built artificially through" with "relative to"

Line 777: replace "... densification and, also, by the... ." with "... densification, and also by the... ."

Line 785: add word "... time series, and fewer... ."

Line 787: replace "To the current... ." with "These current... ."

Line 788: suggest replacing "... is added the... ." with "... is to be considered along with... ."

Line 804: add word "... percentage, and at 100%... ."

Line 807: comment - figure reference (6 (a) and (b)) appears incorrect, and "HR" should by "RH"

Line 809: suggest replacing "Antecedents that press... ." with "The results from this study, along with those of several others, press... ."

Line 812: replace "... time series, that constitutes the database, of two periods... ." with "... time series that constitutes the database of two time periods... ."

Line 817: suggest replacing "Then there is a high probability that  is could be so in many other... ." with "There is a high probability that this could also be shown in many other... ."

Line 825: replace "(T, HR, VV)" with "(T, RH, WV)"

Line 827: replace "in" with "of"

Line 874: add word "... forever, and all current... ."

Line 879: replace "... by an enlarged in short-term... ." with "...by enlarged short-term... ."

Line 895: replace the word "above" with "below"

Line 960: verify Author ("Weizsacker" or "von Weizsacker")

Reviewer 2 Report

The paper analyzes the urbanization process by means of indicators typical of chaos theory. The approach proposed in the paper is innovative and interesting.

As a case study, 6 communes in the metropolitan region of Santiago (Chile) are considered.

The paper addresses a relevant topic, is well organized and generally well written. However, I think that the paper contains some minor issues that have to be better clarified. See my general and specific comments below.

GENERAL COMMENTS

1.The section “Tool for analysis in nonlinear time series” explains the core of the methodology used for the paper. However, the definitions of the parameters provided there are not always rigorous and many of the elements are unclear to me (i.e., what is d_n? Is n in eq 5 the same as in eq 6 and 7? What is the difference between C(m,r) and C(r)?). I suggest carefully reviewing this section being consistent in the different equations.

2.The core of the thesis is the hypothesis that, since the value of some parameters (or indicator of chaoticity, such as Lyapunov exponent, correlation dimension, …) are different in the two time periods 2010-13 and 2017-20, there is a certain trend due to urban densification. My concern is that I’m not sure that two values of a given parameter are enough to determine its trend. The variation could also be generated by the intrinsic randomness of the data. To check whether or not there is a trend in the data, it should be done a different experiment (e.g., computing for each year in a 10-year period the parameters). I’m not asking you to perform new experiments for this paper, but just to highlight the issue mentioned above in the new version of the manuscript if you think that it is effectively a critical point.

3.The computation of the Lyapunov exponent confirms that meteorological variables often exhibit chaotic behaviors. I would point to some recent works that performed the same analysis for wind speed, solar irradiance, ozone concentration.

Fortuna, L., Nunnari, G., and Nunnari, S. “Nonlinear modeling of solar radiation and wind speed time series.” Berlin, Germany, Springer (2016).

Sangiorgio, M., Dercole, F., and Guariso, G. "Forecasting of noisy chaotic systems with deep neural networks." Chaos, Solitons & Fractals 153(2021): 111570.

Sangiorgio, M., Dercole, F., and Guariso, G. "Sensitivity of Chaotic Dynamics Prediction to Observation Noise.” IFAC-PapersOnLine 54.17(2021): 129-134.

4.The paper is quite long (27 pages). I think that the authors could make an additional effort in shortening the length of the paper. For instance, the introduction provides too many specific details that are not essential to frame the work in the right perspective (that should be the scope of an introduction).

The results section presents the same issue: it contains many tables and figures (at least few of them can be deleted) but it lacks some tables/figures that summarize the results.

5.Some of the figures have low quality.

6.In table 2, you present the extended and short name for each station (e.g., La Florida corresponds to EML, Las Condes to EMM, …). I would use the short name in the following, possibly keeping the same order. Otherwise, the reader could find it confusing. (In table 3 you used the extended name, in table 3(a) and table 3(B) it does not make sense using both the names).

SPECIFIC COMMENTS

Line 170-171.I do not understand the sentence “What makes a difference with the previous literature”. It seems that there is a missing part.

Line 247.Is “lineal” correct?

Line 251.Is “linearity” correct? The presence of butterfly effect and chaos requires nonlinearity in the dynamic.

Line 343.The short names EMS and EMO are not presented before. They are introduced in figure 2.

Line 381.I do not understand the sentence “system described by the time series of this study”.

Line 417. Is there a missing part in this sentence?

Line 447.What’s the meaning of BPF?

Line 452.Is “Ministry of the Environmental Ministry” correct?

Line 475.Is “It is also compared to different time maps” correct? I do not understand its meaning.

Line 491.It is not clear what model is validated with Akaike criterion. As far as I know, Kriging is a model while Akaike information criterion is a metric. Can you please clarify this issue?

Figure 2.Axis labels are missing. Since it is a targets vs predictions chart, I would use a 1:1 ration for x and y axis.

Table 3.Remove the top row (La Florida, Las Condes, ….). Remove also the horizontal line under PM2.5.

Equation 8.Use English instead of Spanish.

Line 599.<DeltaI> is not defined.

Table 6.I do not understand the meaning of this table (maybe it can be removed).

Line 612. “for the two study periods”. In table 4, there is only the value for one study period (2017-2020, see line 596).

Figure 3.Use the short names of the stations instead of numbers from 1 to 6 in the x-axis.

Table 7. is “2023” correct? It should be “2013”.

Figure 7 and 8. “Monitoring station” overlaps with “ 0 2 4 6”. Again, I would use the short names of the stations.

Reviewer 3 Report

In this manuscript, the authors aim to quantify the effect of urbanization on the micrometeorological variables. They propose to use Chaotic theory parameters to quantify the changes in two different time periods (2010-2013 and 2017-2020). The work is novel, and results will be definitely helpful to quantify the effect of urbanization on climate parameters. However, the organization of the manuscript is very confusing and not organized. Thus, I suggest that the authors rewrite/reorganize and resubmit the manuscript. 

Major Comments:

  1. The introduction section is too long (9 pages), very verbose, and reads more like a definition. Also, each subsection of the introduction has a summary, data description, and methodology, making the paper hard to read. Every subsection feels disconnected. For example, Lines 253-259 feels like a data description followed by related works. There are also references to most of the figures and tables in the introduction section—for example, Line 353. I recommend rewriting the introduction section to be concise and up to the point so that the motivation and contribution are clear. 
  2. Same issue with the Materials and Methods section as well. Lines 434-444 don't fit the area of study subsection. 
  3. Data section: What are the temporal resolution of the time series? Please include the details in the Data section.
  4. Inconsistency: In lines 482-484, the authors claim that the PM10 time series has no missing data during 2010-2013. However, table 4 shows missing data in PM10. Please check. 
  5. Please check all the figure numbers and table numbers. 
  6. Line 599: How are the values in this table calculated? I could not find the equation for \deltaI.
  7. Figure 4, Figure 7, and Figure 8 have not been referenced in the text.
  8. Figure 5 and Figure 6 are hard to read. 

Detailed Comments:

  1. Line 17: "72 time series": This might mislead the readers because it is a time-series of six parameters at six locations during two time periods. Just "using measurements of meteorological variables ...." will work. 
  2. Line 24: by "built surface" - do the authors mean construction?
  3. Line 33: area of meteorology "that deals with" observations and processes...
  4. Lines 41-42: Can remove "Atmospheric Boundary Layer"
  5. Lines 50-52: Hard to read. 
  6. Line 62, 150: 2-5? Is there a unit?
  7. Line 247: Lineal or Linear?
  8. Lines 287-290: Too long sentence and hard to read. 
  9. Line 342: By m2, do the authors mean area?
  10. Line 347 - 4,6 or 4.6?
  11. Line 383-384: Incomplete sentence
  12. I think the longitude coordinate should be 70.8 W or -70.8E
  13. Line 421: it should be "at 33.5 S and 70.8 E" instead of between. If the area needs to be highlighted, provide "between 33.3-33.65 S and 70.4-70.8 W"
  14. Line 427: What do the authors mean by "in the shade"
  15. Line 429: an "important" number of sources?
  16. Line 430: Which months include this region's fall and winter season?
  17.  Line 463 - I think the authors mean Figure 1 here
  18. Figure 1: The color scheme is very confusing. It seems like the elevation also has a dark color shade. And the population is also highlighted in black color. 
  19. Figure 2: Why is this analysis done? The figure and analysis seem disconnected. 
  20. Line 497: It should be Table-4
  21. Table 4: To avoid multiple rows in the table, the % values can be presented in the parenthesis adjacent to the values.
  22. Line 515: What are dn and d0?
  23. Please rewrite Eqn. 8
  24. Line 595: It should be Table 5. Please change all the figure numbers and table numbers.
  25.  Figure 3: It would be better if the station names (or abbreviations) instead of the numbers. 
  26. Line 716: It should be PM
  27. Line 764, 765 - Incomplete sentence. 

Round 2

Reviewer 3 Report

The new version definitely reads better than the old version. I have some minor comments:

  1. I acknowledge the efforts by the authors in introducing a new "Theoretical Perspective" section. However, I still feel the introduction is not up to the standards of MDPI journals. It would be better if the authors could edit the section.
  2. There were some minor typos and grammatical errors. 
    • Line 451: "Krigeaje"?
    • Line 702: Please check the subscript. It should be 2010-2013
    • Line 716-719: grammatical error
